# The Blood Cytokine Profile of Young People with Early Ischemic Heart Disease Comorbid with Abdominal Obesity

**DOI:** 10.3390/jpm10030087

**Published:** 2020-08-13

**Authors:** Yulia I. Ragino, Veronika I. Oblaukhova, Yana V. Polonskaya, Natalya A. Kuzminykh, Liliya V. Shcherbakova, Elena V. Kashtanova

**Affiliations:** Research Institute of Internal and Preventive Medicine—Branch of the Institute of Cytology and Genetics, Siberian Branch of Russian Academy of Sciences (IIPM–Branch of IC&G SB RAS), 175/1 B. Bogatkova Str., 630089 Novosibirsk, Russia; ragino@mail.ru (Y.I.R.); polonskayayv@bionet.nsc.ru (Y.V.P.); tina87@inbox.ru (N.A.K.); 9584792@mail.ru (L.V.S.); elekastanova@yandex.ru (E.V.K.)

**Keywords:** early IHD, abdominal obesity, blood cytokines/chemokines complex, multiplex assay, Flt3 ligand, GM-CSF, MCP-1, IL-4

## Abstract

Objective: The aim was to study the blood cytokine/chemokine profile of 25–44-year-old people with early ischemic heart disease (IHD) comorbid with abdominal obesity (AO). Methods: A cross-sectional medical examination of subjects in Novosibirsk, Russia, was conducted after random sampling of the above age group. A total of 1457 subjects, 804 females and 653 males, were analyzed. The epidemiological diagnosis of IHD was made in accordance with 17 validated and functional criteria, employing exercise ECG for confirmation. Simultaneous quantitative analyses of 41 cytokines/chemokines in blood serum were performed by a multiplex assay using the HCYTMAG-60K-PX41 panel (MILLIPLEX MAP) on a Luminex 20 MAGPIX flow cytometer, with additional ELISA testing. Results: Flt3 ligand, GM-CSF, and MCP-1 were significantly associated with the relative risk of early IHD. In the presence of AO, GM-CSF, MCP-1 and IL-4 also significantly correlated with the relative risk of early IHD. By univariate regression analysis, the relative risk of early IHD was associated with lowered blood concentrations of Flt3 ligand, whereas the relative risk of early IHD in the presence of AO was associated with lowered blood concentrations of GM-CSF. Employing multivariable regression analysis, only lower blood levels of Flt3 ligand were associated with a relative risk of early IHD, whereas the relative risk of early IHD in the presence of AO was limited to lower levels of IL-4. Conclusion: Findings related to Flt3 ligand, GM-CSF, and IL-4 are consistent with the international literature. Results from the present study are partly confirmative and partly hypothesis generating.

## 1. Introduction

Despite substantial progress in the diagnosis and treatment of ischemic heart disease (IHD), the prevalence of cardiovascular diseases (CVDs) among young people is steadily growing worldwide [1]. The main reason is the increasing prevalence of risk factors of CVDs. One of these risk factors is excess body weight, especially abdominal obesity (AO) [2,3].

The study of a wide range of biomolecules secreted by visceral adipocytes in AO and their effects is an important area in modern fundamental endocrinology. Much attention is given to the notion that AO causes a chronic, systemic, low-intensity, inflammatory process resulting from a combination of increased insulin resistance and higher production of inflammation mediators because of the expansion of the visceral/abdominaladipocyte pool [4,5]. Cytokines, which are endogenous, biologically active mediators of inflammation that are secreted by visceral adipocytes, regulate interactions between cells and between organ systems and determine cell survival, stimulation or suppression of cell growth, and cell differentiation and functional activity, as well as apoptotic processes. Cytokines ensure coordinated actions of immune, endocrine, and nervous systems under normal conditions and in response to pathological factors [6]. It is known that hypertrophied adipocytes, just likelymphocytes and macrophages, produce cytokines and participate in complement activation by launching a chain of inflammatory processes, with the inflammation becoming persistent and systemic. Many studies indicate that a cytokine imbalance is strongly linked with higher risks of cardiometabolic diseases and their complications [7,8,9,10].

Considering the influence of visceral obesity on human health in general, and on IHD in particular, our aim was to investigate the blood cytokine/chemokine profile of young people with early IHD comorbid with AO.

## 2. Materials and Methods

During 2014–2015, a cross-sectional populational medical examination was performed with random sampling of a young population in a typical borough of Novosibirsk, the capital of Western Siberia. The study protocol was authorized by a local ethical committee of IIPM—Branch of IC&G SB RAS (Minutes No. 9 dated 25 June 2013). To compile the study population, we employed a database of the Novosibirsk Territorial Fund of mandatory health insurance, from which 2500 people of both sexes at age 25–44 years were chosen by means of a random number generator. At our Screening Center, 1457 people underwent the medical examination: 804 females and 653 males. All participants gave informed consent to the medical examination and the use of personal data.

The screening was conducted by a team of physicians that had been trained in standardized epidemiological methods of screenings based on medical examination. The medicalexamination program involved the collection of social and demographic data, a socioeconomic questionnaire, questions about cigarettesmoking and alcoholdrinking habits, a nutritional survey, collection of a medication and chronicdisease history, the cardiological Rose questionnaire, three-time measurement of arterial blood pressure (BP), anthropometricdata collection, ECG recording with interpretation in accordance with the Minnesota Code (MC), spirometry, and assessment of other parameters.

The epidemiological diagnosis of IHD was made via validated epidemiological (using the Rose questionnaire) and functional (ECG recording with interpretation via the MC) criteria. The Rose questionnaire is widely used in epidemiological studies as a standardized method for assessing angina. Although it is a screening tool, rather than a diagnostic test, and was originally designed for use in men, it has been found to predict major coronary events in middle aged men and coronary heart disease mortality in both women and men. The diagnosis of “definite IHD” was made if the following criteria were present: a history of large-focal myocardial infarction (ECG with MC), ischemic changes on ECG without left ventricular hypertrophy (ECG with MC), tension angina pectoris (Rose questionnaire), and irregular rhythm and conductance (ECG with MC). Furthermore, the diagnosis of “definite IHD” was confirmed by conducting an ECG test with physical activity (stress test).

Based on diagnosed early IHD, 4 subgroups were formed from the study population (143 subjects total, Table 1): (1)patients with IHD comorbid with AO, 24 subjects;(2)patients with IHD without AO, 25 subjects;(3)age- and sex-matched controls without IHD but with AO, 44 subjects;(4)age- and sex-matched controls without IHD and without AO, 50 subjects.

Blood for biochemical analyses was collected from all subjects from the medial cubital vein in the morning after a ≥12 h fast on the day of the examination. Simultaneous quantitative analyses of cytokines/chemokines in blood serum were performed by a multiplex assay using the HCYTMAG-60K-PX41 panel (MILLIPLEX MAP) on a Luminex MAGPIX flow cytometer. This panel includes quantitation of the following 41 human cytokines/chemokines:
eotaxin (CCL11), epidermal growth factor (EGF), fibroblast growth factor 2 (FGF-2, also known as FGF-basic), fractalkine (CX3CL1), granulocyte colony-stimulating factor (G-CSF), granulocyte-macrophage colony-stimulating factor (GM-CSF), growth-regulated oncogene α (GROα), interferonγ(IFNγ), interferonα2 (IFNα2), interferon γ–induced protein 10(IP-10, also known as CXCL10), interleukin 1 receptor antagonist (IL-1Ra), interleukin 1α (IL-1α), interleukin 1β(IL-1β), interleukin 10 (IL-10), interleukin 12 (IL-12 (p40)), interleukin 12 (IL-12 (p70)), interleukin 13 (IL-13), interleukin 15 (IL-15), interleukin 17A (IL-17A, also known as CTLA8), interleukin 2 (IL-2), interleukin 3 (IL-3), interleukin 4 (IL-4), interleukin 5 (IL-5), interleukin 6 (IL-6), interleukin 7 (IL-7), interleukin 8 (IL-8, also known as CXCL8), interleukin 9 (IL-9), ligand of Fms-like tyrosine kinase 3 (Flt3 ligand), macrophage inflammatory protein1α (MIP-1α, also known as CCL3), macrophage inflammatory protein1β (MIP-1β, also known as CCL4), macrophage-derived chemokine (MDC, also known as CCL22), monocyte chemoattractant protein 1(MCP-1, also known as CCL2), monocyte chemoattractant protein 3 (MCP-3, also known as CCL7), platelet-derived growth factor AA (PDGF-AA), platelet-derived growth factor AB (PDGF-AB/BB), regulated upon activation, normal T cell expressed and presumably secretedchemokine (RANTES, also known as CCL5), soluble ligand of CD40 (sCD40L), transforming growth factor α (TGFα), tumor necrosis factor α (TNFα), tumor necrosis factor β (TNFβ, also known as lymphotoxin-α (LTA)), vascular endothelial growth factor A (VEGF-A).

In addition, some cytokines/chemokines were determined in blood serum by ELISA with the following test systems: Flt3 ligand (R&D), IL-7 (RayBiotech), GM-CSF, sCD40L, IL-2, IL-6, MCP-1, i.e., CCL2, MIP-1β, i.e., CCL4 and TNFα (Bender Medsystems) on ELISA analyzer MULTISCAN-II.

Statistical analysis of the data was performed using the SPSS software (version 17) for Windows, with assessment (for each variable) of the median, mean, confidence intervals (CIs), standard deviation, and lower and upper quartiles. Several methods of group comparison were utilized: Wilcoxon’s test, one-way ANOVA with Dunnett’s test for a multigroup comparison, the Mann–Whitney U test for a comparison of medians, odds ratio (OR) calculation in a logistical regression model, calculation of the OR via contingency tables, the t test, and χ2 test. The 95% threshold of statistical significance was chosen.

## 3. Results

We uncovered differences in concentrations of some cytokines/chemokines between young people with early IHD and those without IHD (Table 2).

In subjects with IHD, blood levels were lower for Flt3 ligand (1.4-fold), GM-CSF (1.9-fold), sCD40L (1.8-fold), IL-7 (1.4-fold), IP-10 (1.3-fold), MCP-1 (1.6-fold), MIP-1β (1.2-fold), and TNFα (1.3-fold) as compared with subjects without IHD.

Next, we carried out an identical analysis between groups of young people with or without AO (Table 3).

In subjects with IHD comorbid with AO, blood concentrations were lower for Flt3 ligand (1.5-fold), GM-CSF (2.3-fold), GROα (2.2-fold), and MCP-1 (1.9-fold), whereas IL-2 concentration in the blood was 3.0-fold higher relative to subjects with AO without IHD. 

To confirm the obtained results for statistically significant differences in the values of cytokines/chemokines between the subgroups, we conducted an additional study of these cytokine/chemokines by ELISA (Table 4). 

According to ELISA data, in subjects with IHD, blood levels were lower for Flt3 ligand (1.56-fold), GM-CSF (2.0-fold), sCD40L (1.7-fold), IL-7 (1.5-fold), MCP-1 (1.76-fold), MIP-1β (1.2-fold), and TNFα (1.26-fold) as compared with subjects without IHD.Thus, significant associations were shown between blood serum cytokine/chemokine measurements by multiplex analysis and ELISA.

The results of subsequent univariateand multivariate logistic regression analyses (with adjustment for age and sex) regarding possible correlations of cytokines/chemokines with the relative risk of early IHD in the whole study population are presented in Table 5. 

The univariate regression analysis revealed that the relative risk of early IHD is significantly associated with lower blood levels of such cytokines/chemokines as Flt3 ligand (OR = 0.965, CI 0.942–0.990, *p* = 0.006), GM-CSF (OR = 0.904, CI 0.832–0.982, *p* = 0.017), and MCP-1 (OR = 0.998, CI 0.996–0.999, *p* = 0.007). Results of the multivariate logistic regression analysis showed that the relative risk of early IHD is significantly associated only with lowered blood concentration of Flt3 ligand (OR = 0.969, CI 0.941–0.998, *p* = 0.039).

Then, univariateand multivariate logistic regression analyses (with adjustment for age and sex) were performed to find possible associations between cytokines/chemokines and the relative risk of early IHD in the studied young people with AO (Table 6). 

According to the univariate regression analysis, the relative risk of early IHD among the subjects with AO is significantly associated with lower blood concentrations of such cytokines/chemokines as GM-CSF (OR = 0.873, CI 0.763–0.999, *p* = 0.049) and MCP-1 (OR = 0.997, CI 0.995–0.999, *p* = 0.027). Judging by the results of the multivariate logistic regression analysis, the relative risk of early IHD among subjects with AO is significantly associated only with a lower blood level of IL-4 (OR = 0.979, CI 0.958–0.999, *p* = 0.049).

Thus, we found that lowered blood concentrations of cytokines/chemokines such as Flt3 ligand, GM-CSF, and MCP-1 are significantly associated with the relative risk of early IHD in young people aged 25–44 years.In addition, in young people 25–44 years old with AO, lowered blood concentrations of cytokines/chemokines are significantly associated with the relative risk of early IHD comorbid with AO: GM-CSF, MCP-1, and IL-4.

## 4. Discussion

In the vast majority of cases, IHD, including early IHD, develops during coronary atherosclerosis, which is a chronic inflammatory process. Cytokines and chemokines are the main biomolecules mediating an inflammatory process. On the other hand, visceral/abdominal obesity is also a factor triggering the development of systemic inflammatory changes in the human body [4,5,6,10]. 

In biomedical research, the modern multiplex technology of biochemical assays allows us to evaluate blood concentrations of a large variety of cytokines and chemokines, including those with an unclear role in the pathogenesis of CVDs. Our results were hardly expected. For instance, not a single proinflammatory cytokine—among those known to be related to the pathogenesis of atherosclerosis—manifested a direct association with the risk of early IHD, including early IHD in the presence of AO. A possible reason is the small number of patients with IHD in our study, but we had to limit the study population to young subjects (aged 25–44 years).

Nonetheless, we detected significant inverse correlations of some cytokines/chemokines with the risk of early IHD. For example, we revealed that lowered blood concentrations of cytokines/chemokines such as Flt3 ligand, GM-CSF, and MCP-1 are significantly associated with the relative risk of early IHD in young people aged 25–44 years. Lowered blood concentrations of two of these cytokines/chemokines—GM-CSF and MCP-1—also significantly correlate with the relative risk of early IHD in the presence of AO. Additionally, a lowered blood level of IL-4 was found to be significantly associated with the relative risk of early IHD in the presence of AO.

Our findings related to Flt3 ligand, GM-CSF, and IL-4 are consistentwith the international literature. For instance, Flt3 ligand (i.e., the ligand of Fms-like tyrosine kinase 3) is a cytokine stimulating the proliferation and differentiation of hematopoietic cells through activation of its receptor, Flt3. Besides this, it serves as the main growth factor of dendritic cells. It has been demonstrated that the number of dendritic cells is low in patients with IHD [11,12,13]. I. Van Brussel and coauthors [14] have reported that this phenomenon is linked with the downregulation of Flt3 ligand in patients with IHD. This observation is suggestive of a protective function of this ligand against IHD.

GM-CSF is a polypeptide cytokine belonging to the group of granulocyte-macrophage colony-stimulating factors. It stimulates the growth and differentiation of granulocytes, macrophages, and eosinophils. In response to inflammation mediators (IL-1, IL-6, and/or TNF-α), many types of cells start to express GM-CSF. It takes part in the pathogenesis of atherosclerosis. For example, there is evidence that mice deficient in GM-CSF are at a substantial risk of atherosclerosis [15]. In a murine experimental model of brain ischemia, GM-CSF reduces the volume of infarction-affected tissue in the brain and enhances the growth of collateral arteries [16]. GM-CSF can have a direct or indirect effect on CVDs by promoting neovascularization of an ischemic myocardium and by alleviating myocardial injury after an infarction [17]. Yiguan Xu and coauthors [18] have revealed that treatment with low doses of GM-CSF (5.0 g/kg) provides a benefit and reduces complications in patients with IHD and that GM-CSF administration at this dose can significantly improve myocardial perfusion and heart function in these patients. Nevertheless, owing to their small sample size, Yiguan Xu et al. stated that further research is needed and that this field holds promise. A study by M. Ditiatkovski and coworkers shows that, in apoE-deficient mice, the volume of atherosclerotic damage and macrophage accumulation increases if GM-CSF is downregulated, suggesting that, in vivo, GM-CSF protects from atherosclerosis [19]. The same authors demonstrated higher expression of adhesive moleculesin atherosclerotic lesions of GM-CSF-deficient mice, also indicating the anti-inflammatory role of GM-CSF in the development of such lesions.

IL-4 is acytokine inducing the differentiation of T helper cells into T helper 2 cells. IL-4 performs a multitude of biological functions, such as stimulation of activated proliferation of B and T cells and the differentiation of the former into plasma cells. This is a key regulator of adaptive (including humoral) immunity. IL-4 is considered an anti-inflammatory cytokine. In a study on a mouse model, Yu. Shintani and coauthors [20] demonstrated the effectiveness of long-acting IL-4 against acute myocardial infarctionin terms of improvements in both systolic and diastolic functions of ventricles.

Our data on the anti-inflammatory chemokine MCP-1 (i.e., CCL2) are hardly consistent with the results of other studies. For instance, it is known that MCP-1 is one of the factors linking obesity-induced inflammation and the development of atherosclerosis and acts by causing macrophage migration into the developing atherosclerotic plaque. Its blood concentration is increased in obese people, thereby recruiting monocytes from bone marrow to tissues via the blood stream [21,22]. MCP-1 can cause division of macrophagic cells in live-tissue implants, whereas MCP-1 deficiency in vivo diminishes the proliferation of adipose-tissue macrophages [23]. Although most of literature data support the involvement of MCP-1 in the pathologies related to obesity, there are some discrepancies. To give an example, K.E. Inouye and coworkers have reported the absence of changes in the number of macrophages in the adipose tissue of MCP-1-deficient mice during high-fat diet-induced obesity [24]. The same authors, however, found that these mice gained more weight and were glucose intolerant [24]. T.L. Cranford et al. have shown that MCP-1 deficiency can differently affect metabolic and inflammatory processes depending on genetic background [25]. A study by Y.W. Lee and colleagues [26] probably can also help to explain our findings about the inverse correlation of MCP-1with the risk of early IHD. These authors reported that IL-4 induces MCP-1 expression in a vascular endothelium, suggesting that this cytokine and this chemokine are unidirectionally linked. We also noted a “unidirectional link”betweenIL-4 and MCP-1, indicatingtheir significant inverse correlations (Table 5) with the relative risk of early IHD in 25–44-year-olds with AO.

In general, our findings related to Flt3 ligand, GM-CSF, and IL-4 are consistent with the international literature. Results from the present study are partly confirmative and partly hypothesis generating.

This study has its own strengths and limitations. The strength of the study is due to the research of a large complex of 41 cytokines/chemokines using a new biochemical technology (multiplex analysis) in young people with early IHD and abdominal obesity. The limitation of the study is the research of only one type of biological material in the examined persons - blood. The collection of other biological material (samples of organs and tissues) from the examined persons was not carried out.

## 5. Conclusions

We identified poorly studied biomolecules inversely correlating with early IHD, including early IHD comorbid with AO: Flt3 ligand, GM-CSF, MCP-1/CCL2, and IL-4. The results undoubtedly require further research on the development of IHD (especially early IHD) in the presence of AO. The use of multiplex biochemical express panels for this purpose should increase the effectiveness of the diagnosis, risk assessment, and prevention of these diseases, with major implications for the young employable population.

## Figures and Tables

**Table 1 jpm-10-00087-t001:** Clinical and anthropometric characteristics of the subgroups under study (median (lower; upper quartile)).

Parameters	Subjects without IHD		Subjects with IHD	
	without AO (n = 50)	with AO (n = 44)	without AO (n = 25)	with AO (n = 24)
Age, years	35.0 (31.0; 40.3)	37.0 (30.5; 42.0)	34.9 (30.7; 41.7)	40.8 (36.0; 45.3)
Systolic BP, mmHg	118.0 (108.0; 126.8)	118.2 (112.1; 130.0)	121.2 (108.3; 133.7)	124.0 (110.5; 148.7)
Diastolic BP, mmHg	77.5 (71.4; 83.4)	80.0 (71.6; 87.5)	79.5 (65.5; 85.7)	80.5 (72.0; 96.6)
Body mass index, kg/m^2^	22.9 (20.8; 25.4)	28.9 (25.6; 33.1)	22.0 (18.9; 24.9)	28.5 (26.0; 34.0)
Heart rate, beats/min	73.5 (64.0; 79.0)	77.0 (68.5; 81.0)	69.5 (63.0; 83.1)	70.5 (65.5; 81.2)
Waist circumference, cm	74.9 (70.5; 76.2)	90.7 (84.0; 98.2)	67.6 (65.3; 78.9)	86.5 (82.8; 97.2)

Note: AO—abdominal obesity, BP—blood pressure.

**Table 2 jpm-10-00087-t002:** Levels of the analyzed human cytokines/chemokines in IHD (median (lower; upper quartile)).

Analytes, pg/mL	Subjects without IHD (n = 94)	Subjects with IHD (n = 49)	*p*
EGF	46.93 [30.38; 77.36]	39.56 [24.12; 68.17]	0.247
FGF-2, i.e., FGF-basic	42.94 [32.02; 66.44]	43.12 [28.9; 53.15]	0.677
Eotaxin, i.e., CCL11	112.42 [79.64; 166.2]	102.85 [75.15; 149.87]	0.382
TGFα	4.44 [2.61; 6.49]	3.78 [2.13; 5.65]	0.119
G-CSF	25.62 [9.7; 41.13]	16.68 [7.61; 38.56]	0.278
Flt3 ligand	40.41 [27.76; 55.54]	28.19 [13.07; 41.99]	**0.003**
GM-CSF	8.36 [4.63; 13.73]	4.46 [2.16; 10.02]	**0.004**
Fractalkine, i.e., CX3CL1	31.49 [24.33; 58.85]	25.75 [22.21; 34.09]	0.159
IFNα2	13.2 [7.89; 23.63]	13.18 [7.97; 25.1]	0.897
IFNγ	9.84 [5.47; 14.98]	9.54 [5.92; 16.18]	0.745
GROα	1480.0 [855.35; 2169.0]	956.45 [580.18; 2019.12]	0.212
IL-10	5.28 [2.59; 7.41]	4.75 [1.31; 7.74]	0.564
MCP-3, i.e., CCL7	19.19 [13.13; 22.99]	20.32 [17.00; 26.8]	0.512
IL-12 (p40)	4.16 [2.55; 14.28]	6.23 [3.12; 15.45]	0.187
MDC, i.e., CCL22	717.04 [518.9; 1016.0]	635.85 [355.31; 998.56]	0.345
IL-12 (p70)	4.86 [1.37; 6.62]	5.12 [1.41; 6.22]	0.958
PDGF-AA	3306.0 [2026.0; 5050.0]	3125.9 [1615.15; 4263.45]	0.159
IL-13	9.13 [7.04; 13.54]	8.95 [6.35; 19.85]	0.687
PDGF-AB/BB	17,351.25 [14,278.13; 22,158.08]	19,956.45 [13,125.12; 22,897.12]	0.715
IL-15	2.72 [1.5; 5.96]	2.87 [1.56; 5.49]	0.955
sCD40L	2230 [842.94; 4068]	1250.12 [185.56; 2974.64]	**0.046**
IL-17A, i.e., CTLA8	3.53 [1.49; 5.38]	2.45 [1.28; 6.11]	0.555
IL-1Ra	6.69 [4.09; 27.84]	6.24 [4.08; 36.88]	0.912
IL-1α	7.63 [5.24; 12.21]	9.56 [4.98; 26.87]	0.871
IL-9	4.14 [2.71; 5.42]	5.12 [2.11; 6.06]	0.412
IL-1β	1.57 [0.74; 2.23]	1.31 [0.59; 2.69]	0.649
IL-2	1.35 [0.64; 3.0]	1.68 [0.68; 2.97]	0.735
IL-3	1.63 [0.73; 2.12]	2.22	0.418
IL-4	71.95 [33.78; 115.43]	56.47 [41.11; 78.54]	0.247
IL-5	1.23 [0.67; 1.65]	1.38 [1.0; 2.68]	0.198
IL-6	2.27 [1.3; 3.6]	3.37 [1.36; 7.95]	0.157
IL-7	10.71 [6.18; 15.87]	7.64 [4.32; 12.85]	**0.029**
IL-8	9.75 [6.73; 13.93]	9.0 [5.11; 15.46]	0.512
IP-10, i.e., CXCL10	211.1 [163.16; 292.73]	165.48 [99.87; 269.45]	**0.035**
MCP-1, i.e., CCL2	516.9 [382.71; 672.92]	326.41 [164.11; 610.47]	**0.003**
MIP-1α, i.e., CCL3	6.79 [4.34; 10.1]	6.12 [4.61; 8.99]	0.748
MIP-1β, i.e., CCL4	29.98 [21.85; 38.91]	24.87 [16.54; 31.28]	**0.035**
RANTES, i.e., CCL5	2870.0 [1897.5; 4289.0]	2741.22 [1312.45; 3812.78]	0.299
TNFα	16.59 [11.99; 20.62]	12.95 [7.95; 18.55]	**0.035**
TNFβ, i.e., LTA	9.17 [7.73; 12.04]	9.87 [6.54; 15.87]	0.995
VEGF-A	97.0 [59.26; 138.69]	84.55 [51.56; 138.45]	0.745

**Table 3 jpm-10-00087-t003:** Levels of the analyzed human cytokines/chemokines in IHD comorbid with AO and without AO (median (lower; upper quartile)).

Analytes, pg/mL	Subjects without AO (n = 75)		*p*	Subjects with AO (n = 68)		*p*
	without IHD (n = 50)	with IHD (n = 25)		without IHD (n = 44)	with IHD (n = 24)	
**EGF**	**45.3 [29.5; 70.7]**	42.3 [22.7; 72.5]	0.654	55.3 [32.2; 91.8]	39.5 [25.8; 67.9]	0.314
FGF-2, i.e., FGF-basic	42.1 [31.3; 63.1]	40.2 [28.1; 52.4]	0.658	45.5 [35.4; 76.3]	46.8 [35.4; 66.9]	0.755
Eotaxin, i.e., CCL11	114.3 [91.8; 165.7]	110.1 [71.5; 174.6]	0.611	98.5 [70; 179.1]	96.5 [75.68; 146.8]	0.689
TGFα	4.1 [2.8; 6.4]	4.32 [1.51; 5.87]	0.687	4.8 [2.47; 6.93]	3.12 [2.02; 4.69]	0.956
G-CSF	20.5 [10.0; 40.5]	18.0 [7.12; 33.56]	0.587	27.8 [9.7; 41.6]	16.8 [7.06; 47.88]	0.489
Flt3 ligand	40 [28.6; 54.9]	29.87 [14.03; 44.12]	**0.021**	40.9 [25.1; 58.8]	26.0 [12.9; 41.03]	**0.048**
GM-CSF	7.5 [4.1; 12.8]	4.98 [2.13; 13.13]	0.197	9.51 [5.64; 14.4]	4.06 [2.11; 9.41]	**0.008**
Fractalkine, i.e., CX3CL1	29.35 [24.3; 45.3]	25.9 [22.11: 32.97]	0.566	33.1 [23.3; 63.4]	26.8 [21.8; 49.8]	0.277
IFNα2	12.7 [7.4; 20.7]	10.13 [5.54; 22.79]	0.534	13.2 [7.89; 23.6]	17.9 [11.88; 27.03]	0.499
IFNγ	10.2 [4.9; 15.6]	8.98 [5.12; 16.56]	0.811	9.6 [6.2; 14.9]	10.21 [6.89; 19.34]	0.645
GROα	1297.5 [727; 1728.3]	1512.1 [567.3; 2239.78]	0.677	1738 [1034; 2624]	796.8 [511.23; 1389.45]	**0.015**
IL-10	6.4 [3.9; 8.7]	4.56 [1.31; 7.45	0.278	3.67 [1.61; 5.92]	4.34 [1.01; 8.12]	0.512
MCP-3, i.e., CCL7	19.4 [16.3; 21.9]	21.51 [16.87; 95.46]	0.499	17.7 [12.3; 23.6]	19.5 [16.51; 23.56]	0.469
IL-12 (p40)	3.6 [2.2; 15.7]	9.78 [3.12; 54.84]	0.245	5.1 [3.07; 13.5]	5.47 [2.31; 15.87]	0.745
MDC, i.e., CCL22	672.4 [482.9; 921.1]	604.8 [214.8; 968.4]	0.498	819.7 [557.6; 1080]	647.87 [387.5; 1035.87]	0.389
IL-12 (p70)	5.5 [2.4; 6.8]	4.75 [1.07; 5.98]	0.389	3.15 [1.04; 5.9]	6.07 [1.14; 6.34]	0.359
PDGF-AA	3146 [2021.3; 5052]	2878.5 [1597; 4159.5]	0.311	3509 [2035; 5009.5]	3198.2 [1585; 4112.1]	0.325
IL-13	9.6 [7.0; 13.5]	8.48 [6.12; 44.51]	0.956	8.64 [7.08; 13.59]	8.85 [6.04; 11.88]	0.569
PDGF-AB/BB	16,303 [14,262; 20,616]	16,367 [13,232; 19,987]	0.745	18,111.4 [14,284.5; 22,804.3]	20,145 [12,366; 29,877]	0.355
IL-15	2.1 [1.2; 4.4]	2.54 [1.23; 4.25]	0.469	3.25 [1.97; 7.26]	3.16 [1.41; 6.31]	0.499
sCD40L	2176.5 [1047; 4057]	2123.7 [175.49; 3864.5]	0.379	2394 [421.5;4126]	1125.4 [197.3; 2705.6]	0.087
IL-17A, i.e., CTLA8	3.5 [1.4; 5.3]	3.45 [1.23; 5.95]	0.867	3.09 [1.72; 6.3]	1.95 [1.01; 6.38]	0.345
IL-1Ra	5.4 [4.0; 33.2]	7.23 [4.03; 86.99]	0.697	8.12 [4.58; 27.7]	5.8 [3.99; 30.26]	0.678
IL-1α	8.1 [5.4; 13.7]	9.3 [5.15; 35.89]	0.311	7.15 [5.05; 11.24]	6.45 [4.01; 10.0]	0.789
IL-9	4.1 [2.7; 5.4]	4.24 [2.09; 8.45]	0.922	4.39 [3.53; 5.61]	4.26 [2.01; 5.12]	0.314
IL-1β	1.51 [0.76; 2.11]	1.45 [0.62; 1.76]	0.411	1.66 [0.62; 2.37]	2.03 [0.61; 2.55]	0.801
IL-2	1.65 [0.76; 3.11]	0.86 [0.61; 2.44]	0.071	0.93 [0.63; 2.76]	2.76 [1.65; 3.49]	**0.031**
IL-3	1.69 [0.98; 2.14]	1.71 [1.07; 2.19]	0.645	1.57 [0.55; 2.1]	1.51 [0.55; 2.11]	0.659
IL-4	52.3 [31.9; 125]	59.01 [50.88; 81.78]	0.899	80.3 [36.8; 110.2]	53.63 [38.64; 73.89]	0.089
IL-5	1.26 [0.6; 1.63]	1.31 [0.66; 2.41]	0.376	1.22 [0.78; 1.73]	1.29 [1.11; 1.89]	0.397
IL-6	2.02 [1.29; 3.12]	5.95 [1.87; 8.97]	**0.009**	2.6 [1.45; 4.03]	1.95 [1.02; 6.13]	0.613
IL-7	8.61 [6.02; 15.4]	6.47 [4.35; 11.55]	0.145	11.45 [6.18; 16.8]	8.67 [5.04; 14.23]	0.159
IL-8	9.65 [6.25; 12.9]	9.65 [4.03; 18.95]	0.957	9.75 [6.92; 15.14]	8.89 [5.56; 14.12]	0.379
IP-10, i.e., CXCL10	203.6 [167.88; 335]	177.1 [102.1; 247.56]	0.089	222.4 [141.6; 281.1]	160.12 [99.87; 270.44]	0.369
MCP-1, i.e., CCL2	474.8 [364.9; 646.1]	369.8 [239.5; 660.8]	0.229	593.8 [451.4; 717.3]	311.5 [173.0; 507.0]	**0.005**
MIP-1α, i.e., CCL3	6.97 [4.06; 9.96]	6.54 [4.31; 10.87]	0.961	6.53 [4.34; 1045]	6.12 [4.99; 8.45]	0.611
MIP-1β, i.e., CCL4	28 [21.4; 35.3]	25.46 [16.11; 30.1]	0.201	31.5 [22.5; 41.4]	26.31 [18.01; 36.31]	0.191
RANTES, i.e., CCL5	3021 [1905.8; 3949.5]	2887.2 [1312.4; 4569.2]	0.587	2795 [1735; 5134]	2749.5 [1365.4; 3645.45]	0.545
TNFα	15.9 [13.2; 20.6]	13.8 [7.45; 19.11]	0.197	17.4 [10.8; 20.9]	13.87 [8.01; 19.87]	0.159
TNFβ, i.e., LTA	8.7 [7.2; 12.6]	10.05 [6.45; 40.31]	0.397	9.46 [8.24; 11.67]	8.95 [6.62; 12.07]	0.405
VEGF-A	79.2 [54.7; 133.5]	86.45 [54.87; 202.45]	0.349	111.3 [65.1; 159.7]	72.45 [53.0; 109.48]	0.145

**Table 4 jpm-10-00087-t004:** Levels of the human cytokines/chemokines in IHD (median [lower; upper quartile]), ELISA data.

Analytes, pg/ml	Subjects without IHD (n = 39)	Subjects with IHD (n = 49)	*p*
Flt3 ligand	41.5 [26.5; 56.6]	26.5 [12.3; 40.6]	**0.002**
GM-CSF	9.5 [4.7; 15.8]	4.7 [2.1; 11.8]	**0.004**
sCD40L	2351.6 [854.6; 4178.9]	1348.7 [190.8; 3144.5]	**0.049**
IL-2	1.2 [0.4; 4.1]	1.9 [1.0; 3.7]	0.750
IL-6	3.4 [1.3; 5.1]	4.8 [1.5; 8.6]	0.255
IL-7	11.8 [6.0; 16.7]	8.0 [4.1; 13.5]	**0.035**
MCP-1, i.e., CCL2	550.7 [325.4; 670.6]	312.4 [125.8; 585.5]	**0.005**
MIP-1β, i.e., CCL4	31.4 [22.4; 40.7]	25.0 [15.5; 30.7]	**0.045**
TNFα	16.5 [10.7; 22.4]	13.1 [8.6; 17.8]	**0.048**

**Table 5 jpm-10-00087-t005:** Results of regression analysis regarding correlations of cytokines/chemokines with IHD risk.

Factors	Univariate Analysis			Multivariate Analysis		
	Exp(B)^1^	95.0% CI	*p*	Exp(B)^1^	95.0% CI	*p*
**Flt3 ligand**	**0.965**	**0.942–0.990**	**0.006**	**0.969**	**0.941–0.998**	**0.039**
**GM-CSF**	**0.904**	**0.832–0.982**	**0.017**	0.972	0.896–1.06	0.499
sCD40L	0.999	0.998–1.001	0.075	1.001	0.999–1.003	0.726
IL-7	0.941	0.883–1.003	0.063	0.952	0.88–1.03	0.219
IP-10, i.e., CXCL10	0.999	0.995–1.002	0.378			
**MCP-1**, i.e., **CCL2**	**0.998**	**0.996–0.999**	**0.007**	0.999	0.997–1.001	0.403
MIP-1β, i.e., CCL4	0.969	0.939–1.000	0.051	0.997	0.961–1.034	0.871
TNFα	0.948	0.891–1.008	0.087			

^1^Exp(B): an odds ratio of a predictor.

**Table 6 jpm-10-00087-t006:** Results of regression analysis regarding correlations of cytokines/chemokines with IHD risk in the presence of AO.

Factors	Univariate Analysis			Multivariate Analysis		
	Exp(B)^1^	95.0% CI	*p*	Exp(B)^1^	95.0% CI	*p*
Flt3 ligand	0.971	0.941–1.002	0.062	0.976	0.937–1.016	0.233
**GM-CSF**	**0.873**	**0.763–0.999**	**0.049**	0.967	0.847–1.104	0.620
GROα	0.999	0.998–1.001	0.097			
sCD40L	1.000	0.999–1.001	0.098			
IL-2	1.000	0.845–1.185	0.998			
**IL-4**	0.983	0.966–1.001	0.057	**0.979**	**0.958–0.999**	**0.049**
**MCP-1**, i.e., **CCL2**	**0.997**	**0.995–0.999**	**0.027**	0.998	0.995–1.001	0.275

^1^Exp(B): an odds ratio of a predictor.

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
