# Peer review of "The Blood Cytokine Profile of Young People with Early Ischemic Heart Disease Comorbid with Abdominal Obesity"

_jpm, 2020, doi:10.3390/jpm10030087_

Round 1

Reviewer 1 Report

The author showed a correlation of cytokines with early ischemic heart disease (IHD) comorbid with abdominal obesity (AO). The manuscript is although written fine but adding confirmation experiments will make findings strong: here are my comments:

  1. the author should do some other technique to validate flow findings. I suggest ELISA or western blotting for confirmation.
  2. sample size should increase to make significance large.
  3. the author should do some histochemistry or immunofluorescent technique to see changes in the heart tissues of a few patients in the study.

Hope these comments will make the paper more attractive for reader.

Author Response

On behalf of the entire team of authors, I would like to thank the reviewer for their valuable time and effort on our manuscript, and are grateful for the insightful comments on and valuable improvements to our paper. We have tried to include most of the suggestions you made, which will definitely help improve our manuscript.

Reviewer Comments to the Authors:

Reviewer 1

The author showed a correlation of cytokines with early ischemic heart disease (IHD) comorbid with abdominal obesity (AO). The manuscript is although written fine but adding confirmation experiments will make findings strong: here are my comments:

  • the author should do some other technique to validate flow findings. I suggest ELISA or western blotting for confirmation.

Author response: We included ELISA data in the article to validate flow findings: in the section Materials and methods - lines 140-142, in the section Results - lines 169-178, including new additional table 4.

  • sample size should increase to make significance large.

Author response: We included an additional 17 people with early IHD in the study material. In connection with the new recalculation of all data on early IHD, we made many changes in tables 1-3 and in the text to them. The results obtained did not change in principle, statistically significant differences remained.

  • the author should do some histochemistry or immunofluorescent technique to see changes in the heart tissues of a few patients in the study.

Author response: We made a section at the end of the article - Strength and limitations – lines 285-291, which indicates that, unfortunately, we could not do histochemical or immunofluorescence analyzes in the tissues of the heart, since such material was not collected in our study

In general, thanks to your comments and remarks, we have significantly redesigned the article and it has become much better for readers.

The text was revised by a professional translator (a native English speaker).

Reviewer 2 Report

Thanks for let me review this interesant paper, but I think it could be improved as follows:

  1. Key words: try to find another words, that can be found in Pubmed (MesH) 
  2. Methods: which ethic comitte autorized the study and what date? In table 1, it may be explained the meaning of AO, BP at the leyend. Please correct typographic errors such as "waist" instead of wrist. The reader needs to remind the Rose questionary. I think it is important to register when did the blood extraction take place (relationship with disease state)
  3. Discussion: At line 206 you mentioned the literature but you did not referenced it. It is necesary to write about limitations and strongness of the study.

Author Response

On behalf of the entire team of authors, I would like to thank the reviewer for their valuable time and effort on our manuscript, and are grateful for the insightful comments on and valuable improvements to our paper. We have tried to include most of the suggestions you made, which will definitely help improve our manuscript.

Reviewer Comments to the Authors:

Reviewer 2

Thanks for let me review this interesant paper, but I think it could be improved as follows:

  • Key words: try to find another words, that can be found in Pubmed (MesH) 

Author response: We changed keywords - lines 29-30.

  • Methods: which ethic comitte autorized the study and what date? In table 1, it may be explained the meaning of AO, BP at the leyend. Please correct typographic errors such as "waist" instead of wrist. The reader needs to remind the Rose questionary. I think it is important to register when did the blood extraction take place (relationship with disease state)

Author response: We indicated the data on the ethics committee - lines 58-59, explained the abbreviations in Table 1 - line 74, corrected errors, indicated information about the Rose questionnaire - lines 76-80, indicated information about blood sampling - line 93.

  • Discussion: At line 206 you mentioned the literature but you did not referenced it. It is necesary to write about limitations and strongness of the study.

Author response: After line 206 (in the new version of the article this is line 227), the literature about which we are talking begins - lines 228-258. We added a section to the article - Strength and limitations – lines 285-291.

In general, thanks to your comments and remarks, we have significantly redesigned the article and it has become much better for readers.

The text was revised by a professional translator (a native English speaker).